# Fibroblasts Adhesion to Laser-Modified Titanium Surfaces—A Systematic Review

**DOI:** 10.3390/ma14237305

**Published:** 2021-11-29

**Authors:** Julia Kensy, Maciej Dobrzyński, Rafał Wiench, Kinga Grzech-Leśniak, Jacek Matys

**Affiliations:** 1Student Scientific Circle of Experimental Dentistry and Biomaterial Research, Faculty of Dentistry, Wroclaw Medical University, Bujwida 44, 50-345 Wroclaw, Poland; julia.kensy@student.umed.wroc.pl; 2Department of Pediatric Dentistry and Preclinical Dentistry, Wroclaw Medical University, Krakowska 26, 50-425 Wroclaw, Poland; 3Department of Periodontal Diseases and Oral Mucosa Diseases, Faculty of Medical Sciences in Zabrze, Medical University of Silesia, Traugutta sq. 2, 41-800 Zabrze, Poland; rwiench@sum.edu.pl; 4Department of Periodontics, School of Dentistry, Virginia Commonwealth University, Richmond, VA 23284, USA; kinga.grzech-lesniak@umed.wroc.pl; 5Laser Laboratory at Dental Surgery Department, Wroclaw Medical University, Krakowska 26, 50-425 Wroclaw, Poland

**Keywords:** CO_2_ laser, discs, erbium laser, implant, neodymium laser

## Abstract

Objective: Laser treatment has been recently introduced in many fields of implant dentistry. The systematic review tried to address the question: “How does laser modification of titanium surface influence fibroblast adhesion?”. Methods: An electronic search of the PubMed and Scopus databases was performed. The following keywords were used: (laser) AND (fibroblast) AND (titanium) AND (implant OR disc) AND (proliferation OR adhesion). Initially, 136 studies were found. Ten studies met the inclusion criteria and were included in the review. All studies chosen to be included in the review were considered to have a low risk of bias. Results: Studies included in the review varied with laser parameters or ways of observing fibroblast behavior. Studies showed that fibroblasts tend to take different shapes and create extensions on modified surfaces and that their metabolic activity is more intense. One study concentrated on laser application and showed that three-directional laser application is the most successful in terms of fibroblast adhesion. Studies which concentrated more on laser parameters showed that too low energy density (lower or equal to 0.75 J/cm^2^) does not influence fibroblast adhesion. Increasing the energy density over 0.75 J/cm^2^ causes better cell adhesion of fibroblasts to the laser-modified sample. One included study focused on increasing titanium surface wettability, which also positively influenced cell adhesion. Conclusion: The studies included in the review proved a positive effect of laser-modified titanium surfaces on fibroblast adhesion. However, the application of an appropriate laser energy dose is crucial.

## 1. Introduction

In recent decades, dental implants have become common prosthetic rehabilitation for missing teeth [1,2]. They effectively improved other prosthetic rehabilitation methods such as partial and complete dentures or teeth-supported prosthetic reconstruction [3,4,5]. This phenomenon is because the implant’s esthetic features and the comfort of using it are superior to the traditional prostheses [3]. However, dental implant rehabilitation is more expensive and demands surgical intervention [1] Moreover, the survival rate in compromised patients (diabetes, elderly) decreased thus the implants with the surface pronounced the bone osseointegration are necessary [6,7]. To eliminate potential negative consequences researchers have been searching for optimal methods and biomaterials to provide a high contact of the implant to the bone.

A first-choice material in dental implants is titanium [8] It exhibits properties such as resistance to corrosion by fluids and acids produced by the human body or by oxygen, light weight, strength (titanium can withstand the forces of chewing) and biocompatibility [8] Titanium and its alloys are well tolerated by surrounding living tissue which promotes osseointegration around the implant [9]. However good bone regeneration is not enough for the successful treatment. In subsequent years after prosthetic reconstruction supported on implants, titanium is surrounded with peri-implant mucosal tissue which includes epithelial cells and fibroblasts [10]. The growing tissue creates a narrow connection between the material and the surrounding tissue [11]. This phenomenon protects from bacterial penetration into the sulcus which can lead to peri-implatitis and implant loss [12]. Titanium can be easily modified by varied treatments (chemical or mechanical) such as laser treatment, sandblasting, or acid etching which increases its beneficial features [2,6]. Both properties of titanium (micro and macroscopic) and quantity and quality of the bone are responsible for the treatment success [13].

Surface modification of the implant stimulates the growth of the fibroblasts which improves the healing and implantation process [12,14]. Cell adhesion to the implant surface is determined by the immune system response which consists of creating a thin layer of cells such as thrombocyte [15]. Due to contact with water contained in physiological fluids a layer of specific proteins which can affect the abundant surface (ex. integrins, cadherines, etc.) is created [16]. The proteins stimulate epithelial cells and fibroblast cells to adhere to the implant. Once they adhere, they adjust to the surface by changing the shape and growing extensions [17]. Fibroblast adhesion is a crucial phenomenon in successful treatment with dental implants. This capability is responsible for creating a proper gingival attachment between soft tissue and titanium implant surface [18].

Laser application in implant dentistry has been widely introduced [19,20,21,22,23]. Laser therapy found usage broadly in periodontology and surgery but also in endodontic treatment or cavity preparation [24,25,26,27,28,29,30,31]. Laser irradiation can be used to modify the surface roughness changing its topography. Scanning electron microscope (SEM) analysis shows a huge difference of the implant structure after the laser modification. Non-treated samples are smooth, while laser treatment generates grooves and pores. This therapy seems to increase the adhesive area and increase surface wettability and abutment integration and the chance of successful implant treatment [32].

Depending on the stage of peri-implantitis, various lasers can be applied to remove debris and granulation tissue covering the implant tissue at different levels (collar part, mid-high level). The use of lasers with high power can change the structure of the implant surface [32]. From a clinical point of view, it is essential to answer the question; does titanium surface which was changed (modified) with lasers have a high ability for the fibroblasts adhesion to the implant surface, which is the crucial factor responding for the fibroblasts’ adhesion to the implant surface and gingival peri-implant pocket reduction.

The systematic review aimed to investigate how titanium surface modification using different wavelengths influences fibroblast behavior (proliferation) and whether the laser application can increase the implant to soft tissue contact.

## 2. Materials and Methods

### 2.1. Protocol and Focused Question

A detailed description of text selection in the review was structured in accordance with PRISMA Statement [33]. The protocol is included in the paper (Figure 1).

Focused question:

In this paper, the researchers focused on a question—“How laser modification of titanium surface influences fibroblast adhesion?”

### 2.2. Eligibility Criteria

Only studies that met the criteria below were included in the review:detailed laser conditioning—specified type and parameterstitanium samples (titanium alloys were also included)fibroblast cells used as proliferating cellsin vitro studiesstudies with a control groupstudies in EnglishReviewers agreed to exclude the following criteria:studies without a control groupsamples not made of pure titanium or its alloysnon-English studiesclinical reportsreview articlesreviewsmeta-analysis

No restrictions were applied regarding the year of publication.

### 2.3. Information Sources, Search Strategy, and Study Selection

The literature review in PubMed and Scopus databases was conducted in September 2021 to find articles that related to laser’s influence on fibroblast adhesion to titanium surface. A specific search term (laser) AND (fibroblast) AND (titanium) AND (implant OR disc) AND (proliferation OR adhesion) was applied to find. The reviewers limited the search only to in vitro studies that related to eligibility criteria. Inclusion criteria consisted of in vitro studies where a titanium material was conditioned with a laser treatment. Only English studies and the ones available in full-text version were included. Criteria such as no laser conditioning, non-titanium material used or non-English papers were excluded. Meta-analysis and other review articles were not scanned.

### 2.4. Data Collection Process, Data Items

Data from papers that met the inclusion criteria was extracted by the two reviewers independently. The following data were used: first author, year of publication, study design, article title, laser type and specific changes in cell adhesion before and after laser modification and results. Extracted data were enrolled into an Excel spreadsheet created for this research.

### 2.5. Risk of Bias in Individual Studies

At the preliminary study selection each reviewer screened titles and abstracts separately to minimize the potential bias Cohen k test was used as a tool to determine the level of agreement between researchers [34]. In the case of any difference in opinion on the study inclusion or exclusion was resolved by discussion between the authors.

### 2.6. Quality Assessment

Two blinded reviewers screened the studies separately and independently to evaluate the quality of each included study. To establish study design, implementation and analysis the following criteria was used: sample quantity, indicated incubation time, full titanium sample specification, surface characteristics after laser application, fibroblast cells origin, laser type and laser parameters. The studies were graded on a scale from 0 to 9 points. The higher score indicated higher quality of the study. Any disagreements were resolved through discussion until reaching consensus.

### 2.7. Risk of Bias across Studies

The scores of each study were calculated and an overall estimate risk of bias (low, moderate, high) was made for each included study, as recommended in the Cochrane Handbook for Systematic Reviews of Interventions [35].

## 3. Results

### 3.1. Study Selection

The initial database search recognized 136 articles which were potentially applicable for the review. First title and abstract screening allowed to exclude 110 articles as not focused on the reviewed subject (no laser treatment used, other cells than fibroblasts for example bacteria, different sample material). Eighteen studies were selected for full-text screening, from which 8 were excluded due to not meeting defined inclusion criteria [36,37,38,39,40,41,42,43]. Ten papers were included for qualitative synthesis [37,38,39,40,41,42,43,44,45,46]. The flowchart below presents the main reasons for exclusion (Table 1.)

### 3.2. General Characteristics of the Included Studies

Ten studies were included in this review. In each study different lasers were used with different parameters that allowed to compare the influence of these devices on fibroblast adhesion to titanium surfaces. The general characteristics of each study and materials used are presented in Table 2.

In included studies titanium was the only material used; however, in each study the samples’ surface was prepared in a specific way—sandblasted, acid-etched, polished, autoclaved, washed in ultrasonic bath, cleaned with alcohol, acetone or water. Depending on the study the way of abundant decontamination was slightly different. Titanium was treated with different lasers i.e., Er:YAG [49], erbium fiber laser [52], ytterbium fiber laser [53], Ti:Sapphire laser fs-system laser [44,46], Nd:YAG [47], Nd:YVO_4_ [51]_,_ CO_2_ laser [48], KrF excimer laser [45], diode laser [50]. In eleven studies laser was the only treatment applied. In a study by Cao et al. [49] other kinds of surface modification were used to compare the adhesion. The methods applied in Cao et al. [49] study were stainless steel curette modification, ultrasonic system with straight carbon fiber tip and metal tip and rotating titanium brush. However, the elements that did not correspond to the review were omitted.

### 3.3. Subjects of the Study

The full-text articles review found heterogeneity in the papers. All the studies included in the review concerned fibroblasts—whether human cells from biopsies [46,49,50,51] or cell culture [52,53] or mouse embryonic cells [44,47,48]. In one study, cell origin was not defined [45]. In each case the cells were properly prepared to increase adhesion and proliferation features. The most common method was cell supplementation with fetal bovine/calf serum, antibiotic therapy and trypsinization. In a study by Lee DW et al. [46] L-glutamine was applied as an additional supplementation. Similarly in study by Chikarakara et al. [48] amphotericin B was added. The authors of one study did not precise the preparation procedure [45] (Table 3).

The studies varied in kind of laser type and the lasing parameters. The surface modification consisted of creating microgrooves and microroughness on the disc which widened contact surface between the sample and the cells. Thanks to this modification the fibroblasts were prompted to grow and to form pseudopods which could influence better adhesion. The lasers used in included papers could be divided into the following groups: Nd:YAG [47], Nd:YVO_4_ [51], Er:YAG [49], erbium fiber laser [52], ytterbium fiber laser [52], Ti:Sapphire laser [44,46], diode laser [49], KrF excimer laser [45] and CO_2_ laser [48] (Table 4).

### 3.4. Main Study Outcomes

The studies included in the review were not homogenic in terms of laser treatment. They varied with laser wavelength, energy density and power output. Some did not precise any lasing parameters [51]. The researchers decided to have a more general look on the modification and how it influences fibroblast behavior. The main study outcome is that despite differences in laser wavelength and parameters, the cell adhesion increased. It could be concluded by observing fibroblasts’ morphology changes [46,50,51,52,53] or by examining changes in producing proteins responsible for adhesion [46,49,53]. The most common method of examining cell morphology was using SEM [46,50,51,52,53]. Fibroblasts showed various shapes on modified surfaces [50], denser growth [52], generating microstructures such as long extensions (pseudopodia) [44,46,51]. These structures followed the grooves or created bridges if the holey structures were prepared on the samples [45]. Certain studies analyzed the problem in microscale—adherent protein expression was measured [49,53]. Studies concentrated on proteins such as vinculin [46,53], FAK (focal adhesion kinase) [49,51], ITGB1 (integrin-beta1) [49] or integrin-beta4 [46] showed that protein expression was more intense on laser-modified specimens which proves the increased adhesion.

### 3.5. Quality Assessment

The articles included in the review were qualified as high-quality scoring 7/9 [44,46,48,49,53], 8/9 [47,52] or 9/9 points [50]. None of the studies was obtained and excluded as low-quality or moderate paper, but some of the studies were qualified as a moderate risk of bias scoring 6/9 points [45,51] (Table 5). However, missing information was not crucial for the research results.

## 4. Discussion

Most studies that met the inclusion criteria and were considered in the review proved that laser therapy can increase cell adhesion to modified surfaces. The included papers could be divided into seven groups depending on what laser type was used—Nd:YAG [47], Nd:YVO_4_ [51], Er:YAG [49], erbium fiber laser [52], ytterbium fiber laser [53], Ti:Sapphire laser [44,46], diode laser [50], KrF excimer laser [45] and CO_2_ laser [48]. All these applied lasers have ability to alter the surface of the titanium which promoted the adhesion of the fibroblasts regardless of the type of fibroblast line and the grade of titanium alloy.

In studies by Vignesh et al. [47] and by Baltriukiene D et al. [51] authors used different neodynium lasers—sNd:YAG [54] and Nd:YVO_4_ [55]—to modify titanium discs. However, the results reached in the studies were similar. After SEM analysis in both cases a specific surface landscape was observed presenting characteristic roughness similar to troughed and holey structures [47,51]. Moreover, laser treatment keeps the purity of the disc, not contaminating it with additional compounds such as by-products of mechanical treatment [47]. In each study different fibroblasts cells were used—from biopsies [51] or from cell culture [47]; however, the result of these two studies was the same. Cells were observed under the SEM minimum 24 h after seeding. Low magnification showed a huge variety of cell shapes and extensions they grew. More careful observation at high magnification showed that the extensions grow into microgrooves and pores created by laser beam. Similar results were reached in other included studies [44,52] where SEM analysis proved the presence of pseudopodia and lamellipodia in laser-dimpled areas. All papers reported better cell adhesion to modified Ti-implant.

Certain studies examined cell viability by measuring cell metabolic activity. In their study Chikarakara et al. [48] showed that roughness induces cells activity as the percentage of reduced resazurin was higher. Studies conducted by Baltriukiene et al. [51] and Khadra et al. [50] also examined cell viability. However, in these cases Nd:YVO_4_ and diode laser treatment do not seem to increase this feature as the viability was the same in all samples.

Er:YAG laser was used in the study by Cao J et al. [49]. This study did not concentrate on cell morphology but considered gene expression responsible for adhesion (FAK and ITGB1) and ECM synthesis (COL1A1 and FN1). Results showed that all treatments applied in the study increased HGF’s adhesive strength but only on sandblasted surfaces. Samples cleaned only with acetone and alcohol showed reduction of the strength. The SAE (sandblasted) samples showed better conditions for gene expression, especially on SAE and laser-modified surfaces where a significant increase of COL1A1 expression was noticed. However, the positive results of laser treatment were only observed in SAE samples. Laser treatment did not increase surface roughness parameters in the case of other samples [49].

Study based on erbium fiber laser use proved that cell attachment depends on topography itself [52]. That means that the shape, direction and crosshatching of the grooves is crucial for good adhesion. The best evidence for that is the study where each group was laser-modified in different ways—unidirectional application, two-directional and three-directional crosshatching. Three-directional modification was the most successful titanium treatment in the context of fibroblast adhesion. Thus, the crosshatching not only increased cell adhesion but also contributed to better fibroblast spread on the examined surface [52].

The laser type seems not to be crucial in enhancing cell response. More important are the parameters and dosage. Laser energy density must be set for particular parameters, because too low energy density will not cause any difference in cell reaction compared to the control sample [50]. After reaching the lower limit, the increase of the energy dose influences surface roughness (the distance between the nanostructures is bigger) which entails better cell growth (longer extensions) [44]. Similar reaction is going to be reached by exposing the sample to multiple laser doses instead of one dose [45]. Multiple doses or increasing the number of pulses increase surface roughness which causes better cell adhesion however the difference in cell adhesion can be only noticed compared to control trials [45,50].

Except for the parameters described above, some studies considered the difference of wettability of the control and treated samples [44,53] cases showed that laser therapy increases surface wettability as reducing contact (wettability) angle. Therefore, the contact of the titanium implant with physiological fluids is better and protein adsorption increases what is a key to successful cell adhesion to the abundant and fast peri-implant tissue healing process [46]. However, some laser therapies can reduce contact angle too much (below 10 grades). This effect can lead to cell adhesion disorders or unsuccessful alignment what results in implantation treatment failure [53].

## 5. Conclusions

From the included studies, it can be concluded that ytterbium fiber laser, erbium fiber laser, Nd:YVO_4_, Er:YAG, CO_2_, Nd:YAG, KrF excimer laser, GaAlAs diode laser and Ti:Sapphire laser therapy positively influences fibroblast cells adhesion to the modified titanium surface. The review shows that most of laser type treatment increases surface roughness stimulating the cells to adhere and proliferate. However crucial is laser density and multiple exposures.

## Figures and Tables

**Figure 1 materials-14-07305-f001:**
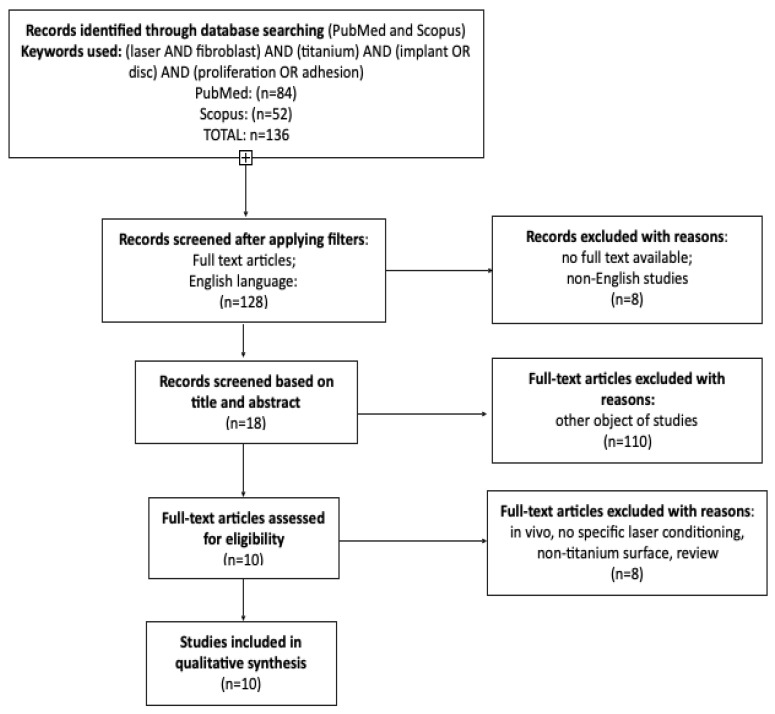
PRISMA flowchart presenting the criteria for the included studies.

**Table 1 materials-14-07305-t001:** Reasons for exclusion of studies.

Ordinal Number	Reason	Reference Number
1	no pure titanium discs, laser type not specified	Schaeske J [36]
2	in vivo study, laser type not specified	Lee HJ [37]
3	review	Corvino E [38]
4	no pure titanium discs	Zhang Q [39]
5	in vivo study, osteoblasts	Khosroshahi ME [40]
6	in vivo study	Abrahamsson I [41]
7	laser type not specified	Perez-Diaz L [42]
8	laser type not specified	Gheisarifar M [43]

**Table 2 materials-14-07305-t002:** General characteristic of the included studies.

Ordinal Number	Type of Sample	Material and Preparation	Laser Type and Parameters	Number of Samples	Reference Number
1	Discs of 10 mm diameter	99.99% pure titanium, grinded and polished	Ti: Sapphire fs-laser system	Not specified	Aliuos P [44]
2	15 mm implants of diameter 4.5 mm	Pure titanium, plasma sprayed	KrF excimer laser	8	Heinrich A [45]
3	Discs 10 mm × 2 mm	Pure, grade 4 Ti plates	Ti: Sapphire fs-laser system	48	Lee DW [46]
4	Discs 6 mm × 2.3 mm	CpTi grade II, sandblasted	Nd:YAG	40	Vignesh [47]
5	Workpiece 100 mm × 20 mm × 4 mm	Ti-6Al-4V alloy grid blasted	CO_2_ laser	4 workpieces, each divided into 10 sections	Chikarakara E [48]
6	Discs 15 mm × 1 mm	Pure titanium one group mechanically polished and ultrasonically cleaned with pure acetone and ethanol (M), second group additionally sandblasted (SAE)	Er:YAG	12	Cao J [49]
7	Discs 10 mm × 1 mm	Grade 4 pure titanium, polished, treated with trichloroethylene, rinsed with ethanol, treated with ethanol bath, autoclaved	GaAlAs diode laser	40	Khadra M [50]
8	Discs of 5.2 mm × 2 mm	Grade 2 titanium alloy, sterilized, exposed to UV light	Nd:YVO_4_	25	Baltriukiene D [51]
9	Discs 16 mm × 2 mm	Grade 4 pure titanium, cleaned with distilled water	Erbium fiber laser	28	Çelebi H [52]
10	Discs 10 mm × 3 mm	Ti-6Al-4V alloy, polished, washed in acetone	Ytterbium fiber laser	Not specified	Aktas OC [53]

**Table 3 materials-14-07305-t003:** Characteristics of cells used in the studies.

Ordinal Number	Cell Line	Preparation	Reference Number
1	NIH 3T3 fibroblasts	Supplemented with fetal calf serum, penicillin/ streptomycin; trypsinized	Aliuos P [44]
2	Fibroblasts cells	N/A	Heinrich A [45]
3	Human lower gingival epithelial squamous carcinoma cell line YD-38 Human fetal lung fibroblast-like cell line MRC-5	Supplemented with fetal bovine serum, L-glutamine, penicillin/streptomycin	Lee DW [46]
4	L929 murine fibroblasts	Supplemented with fetal bovine serum, penicillin/streptomycin; trypsinized	Vignesh [47]
5	BALB 3T3 and NIH 3T3 mouse embryonic fibroblast cell line	Supplemented with fetal calf serum, penicillin, streptomycin and amphotericin B,	Chikarakara E [48]
6	Human gingival fibroblasts from biopsies	Supplemented with fetal bovine serum, penicillin/streptomycin	Cao J [49]
7	Human gingival fibroblasts from biopsies	Supplemented with fetal calf serum, penicillin/streptomycin	Khadra M [50]
8	Human gingival subepithelial cells from biopsies	Supplemented with fetal bovine serum, penicillin/streptomycin	Baltriukiene D [51]
9	Human gingival fibroblasts (HGF-1) from cell culture	Supplemented with fetal calf serum, penicillin/streptomycin	Çelebi H [52]
10	Primary human fibroblasts (HGFIBs) from cell culture	growth medium (FCS), trypsinized	Aktas OC [53]

**Table 4 materials-14-07305-t004:** Characteristics of lasers used for treatments.

Ordinal Number	Laser Type	Wavelength (nm)	Energy Density (J/cm^2^)	Power Output (mW)	Reference Number
1	Ti:Sapphire fs-system laser	800	20 J/cm^2^ 100 J/cm^2^	N/A	Aliuos P [44]
2	KrF excimer laser	248	~15–17 J/cm^2^	N/A	Heinrich A [45]
3	Ti:Sapphire fs-system laser	800	N/A	5 mW	Lee DW [46]
4	Nd:YAG	N/A	1.5–4.5 J/cm^2^	4 kW	Vignesh [47]
5	CO_2_ laser	N/A	N/A	1.5 kW	Chikarakara E [48]
6	Er:YAG	N/A	30 mJ/pulse	N/A	Cao J [49]
7	GaAlAs diode laser	830	1.5, 3 J/cm^2^	84 mW	Khadra M [50]
8	Nd:YVO_4_	N/A	N/A	N/A	Baltriukiene D [51]
9	Erbium fiber laser	N/A	0.1–0.5 mJ/pulse	20 W	Çelebi H [52]
10	Ytterbium Fiber Laser	1085	N/A	25 W	Aktas OC [53]

**Table 5 materials-14-07305-t005:** Quality assessment of the included studies.

Criteria	First Author									
	Aliuos et al. [44]	Heinrich et al. [45]	Lee et al. [46]	Vignesh et al. [47]	Chikarakara et al. [48]	Cao et al. [49]	Khadra et al. [50]	Baltriukiene et al. [51]	Çelebi et al. [52]	Aktas et al. [53]
Sample quantity	0	0	1	1	1	1	1	1	1	0
Indicated incubation time	1	0	1	1	1	1	1	1	1	1
Titanium discs full specification	1	1	1	1	1	1	1	1	1	1
Surface specification after preparation	1	1	0	1	1	1	1	1	1	1
Fibroblast cells origin	1	1	1	1	1	1	1	1	1	1
Laser type	1	1	1	1	1	1	1	1	1	1
Laser parameters:										
Wavelength	1	1	1	0	0	0	1	0	0	1
Power output	1	1	0	1	0	1	1	0	1	0
Energy density	0	0	1	1	1	0	1	0	1	1
Total	7	6	7	8	7	7	9	6	8	7
Risk of bias	Low	Moderate	Low	Low	Low	Low	Low	Moderate	Low	Low

## Data Availability

The data presented in this study are available on request from the corresponding authors.

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
