# Peer review of "Fibroblasts Adhesion to Laser-Modified Titanium Surfaces—A Systematic Review"

_materials, 2021, doi:10.3390/ma14237305_

Round 1

Reviewer 1 Report

The manuscript by Kensy et al. reports a systematic review focusing on whether laser-treated titanium surface can enhance fibroblast adhesion. Laser ablation is a surface-modifying technique used to create micro topography on the titanium implant collar, which has been proposed to elicit the attachment of connective tissue and bone and inhibit epithelial downgrowth. Some beneficial outcomes of the laser microtexturing have been reported in preclinical and clinical studies indicating improved soft tissue attachment and crestal bone preservation. However, the available evidence is highly variable and far from conclusive. Although the current manuscript focuses on a topic related to an important clinical question, there are conceptual and methodological issues which need to be addressed.

Major comments:

  1. It is unclear why the fibroblast adhesion was used as the primary outcome in this systematic review, and that only in vitro studies were included while all the in vivo/clinical studies were excluded. Is fibroblast adhesion a determinant for the survival of dental implant? In the reviewer’s opinion, how the gingiva tissues response to the laser-modified implant collar may be more clinically relevant and important, as it may create a biological seal which is critical for the peri-implant health. If the authors are more interested in the impact of laser treatment on fibroblast adhesion on dental implants, a narrative review could be more appropriate and more interesting to the readers of the journal.
  2. Final studies included in the synthesis are too few (N=10), which significantly limits the robustness of the review. This may be partially due to the search procedure proposed using keywords only, which is not comprehensive enough to ensure retrieval of the most relevant information. It is recommended to use Medical Subject Headings (MeSH) in addition to keywords in the literature search procedure.
  3. Assessment for heterogeneity in the literatures is lacking. Surface topography, laser type, cell types (primary/cell lines) are highly diverse across studies, which were not properly analyzed. Also, most studies have variable measurements for the outcome (cell adhesion) and observation time point. Simply combining experiment without carefully looking into the variation in outcome measures will lead to wrong cause-effect estimates and consequently to misleading conclusions. It is very hard to define whether there was an actual adhesion improvement. Thus, the conclusion that laser can enhance adhesion, regardless of laser type is not convincing.

Minor comments:

  1. In the Introduction, there is a lack of information on what the current uncertainties are regarding the topic.
  2. It is unclear why the Cochrane Central Register of Controlled Trials (CENTRAL) database was searched, if only in vitro studies were to be included.

Author Response

Dear reviewer, thank you for your great effort in review of our manuscript. The changes are highlighted in blue color.

The manuscript by Kensy et al. reports a systematic review focusing on whether laser-treated titanium surface can enhance fibroblast adhesion. Laser ablation is a surface-modifying technique used to create micro topography on the titanium implant collar, which has been proposed to elicit the attachment of connective tissue and bone and inhibit epithelial downgrowth. Some beneficial outcomes of the laser microtexturing have been reported in preclinical and clinical studies indicating improved soft tissue attachment and crestal bone preservation. However, the available evidence is highly variable and far from conclusive. Although the current manuscript focuses on a topic related to an important clinical question, there are conceptual and methodological issues which need to be addressed.

Major comments:

1. It is unclear why the fibroblast adhesion was used as the primary outcome in this systematic review, and that only in vitro studies were included while all the in vivo/clinical studies were excluded. Is fibroblast adhesion a determinant for the survival of dental implant? In the reviewer’s opinion, how the gingiva tissues response to the laser-modified implant collar may be more clinically relevant and important, as it may create a biological seal which is critical for the peri-implant health. If the authors are more interested in the impact of laser treatment on fibroblast adhesion on dental implants, a narrative review could be more appropriate and more interesting to the readers of the journal.

Ad 1. Appropriate corrections have been introduced to review introduction as suggested by the Reviewer. Only in vitro trials allow to observe cell attachment and allow to compare the results with a control trial.

Depending on the stage of periimplantitis, various lasers can be applied to remove debris and granulation tissue covering the implant tissue at different levels (collar part, mid-high level). The use of lasers with high power can change the structure of the implant surface. From a clinical point of view, it is essential to answer the question; does titanium surface which was changed (modified) with lasers have a high ability for the fibroblasts adhesion to the implant surface, which is the crucial factor responding for the fibroblasts' adhesion to the implant surface and gingival peri-implant pocket reduction. Fibroblast adhesion is a crucial phenomenon in successful treatment with dental implants. This capability is responsible for creating a proper gingival attachment between soft tissue and titanium implant surface. Improper fibroblasts adhesion and gingival attachment can lead to higher bacteria penetration into the gingival pocket around the implant and cause peri-implant inflammation and increase the risk of implant loss.

2. Final studies included in the synthesis are too few (N=10), which significantly limits the robustness of the review. This may be partially due to the search procedure proposed using keywords only, which is not comprehensive enough to ensure retrieval of the most relevant information. It is recommended to use Medical Subject Headings (MeSH) in addition to keywords in the literature search procedure.

Ad 2. We screened also the Scopus database however only 10 articles were included in the review, as only this amount fulfilled all the criteria created by the authors. However, even this amount of studies gives the clinicians and examiners general picture and idea of laser treatment influence on implant integration after implantation procedure. 

Not all words used as keywords can be found in MeSH, but the words are commonly used in literature related to implantology. The words have been used to find as many articles as possible and not to omit any article which could be essential in terms of content.

3. Assessment for heterogeneity in the literatures is lacking. Surface topography, laser type, cell types (primary/cell lines) are highly diverse across studies, which were not properly analyzed. Also, most studies have variable measurements for the outcome (cell adhesion) and observation time point. Simply combining experiment without carefully looking into the variation in outcome measures will lead to wrong cause-effect estimates and consequently to misleading conclusions. It is very hard to define whether there was an actual adhesion improvement. Thus, the conclusion that laser can enhance adhesion, regardless of laser type is not convincing.

Ad 3.  Corrections have been introduced to the Quality assessment and risk of bias paragraph. We added suggested variables to Table 5. Thank you for this important remark.

Minor comments:

1. In the Introduction, there is a lack of information on what the current uncertainties are regarding the topic.

Ad 1. Appropriate corrections have been introduced to review introduction as suggested by the Reviewer.

2. It is unclear why the Cochrane Central Register of Controlled Trials (CENTRAL) database was searched, if only in vitro studies were to be included.

Ad 2.  In this article the Cochrane Central Register of Controlled Trials (CENTRAL) database was not searched. Only the Cochrane Handbook for Systematic Reviews of Interventions was used to estimate risk of bias.

Reviewer 2 Report

In the paper entitled “Fibroblasts Adhesion to Laser Modified Titanium Surfaces. A Systematic Review”, the authors are aiming for a systematic review, but more work needs to be done to improve the paper.

The authors should use more databases to search for papers, not only pubmed.

Although the authors give a flow-chart presenting the criteria regarding which studies are included in the paper, the selection/exclusion criteria should be better stated.

For example. It should be also made clear that only studies on pure titanium discs are included.

Also in table 1 authors are excluding papers mentioning that report in vivo studies (ref 30 and 33), however in both studies are also reported in vitro studies. For ref 33, there are also reports for fibroblasts.

Also, after a rapid search, in pubmed, according to the authors search words, paper which pass the authors criteria were not found between the final 10.

Author Response

Dear reviewer, thank you for your great effort in review of  our manuscript. The changes are highlighted in blue color.

In the paper entitled “Fibroblasts Adhesion to Laser Modified Titanium Surfaces. A Systematic Review”, the authors are aiming for a systematic review, but more work needs to be done to improve the paper.

1. The authors should use more databases to search for papers, not only pubmed.

Ad 1. We added also results from Scopus database. Thank you for this important remark.

2. Although the authors give a flow-chart presenting the criteria regarding which studies are included in the paper, the selection/exclusion criteria should be better stated.

Ad 2. The authors modified 2.2 Eligibility criteria paragraph to present the inclusion/exclusion criteria more clearly.

3. For example. It should be also made clear that only studies on pure titanium discs are included.

Ad 3. The exclusion criteria says that non-titanium samples were not taken into consideration in this article. What is more, Table 2. clearly presents abundant parameters and in each case the discs were made of pure titanium or Ti-6Al-4V alloy.

4. Also in table 1 authors are excluding papers mentioning that report in vivo studies (ref 30 and 33), however in both studies are also reported in vitro studies. For ref 33, there are also reports for fibroblasts.

Ad 4. Both articles were excluded, because the authors wanted to fully concentrate on in vitro studies. Any comparisons between in vitro and in vivo results were eliminated. Furthermore, ref 30 (now ref 37 as the reference numbers have changed) do not include laser therapy.

5. Also, after a rapid search, in pubmed, according to the authors search words, paper which pass the authors criteria were not found between the final 10.

Ad 5.  The PubMed and Scopus search using the keywords presented in the article shows more results, but after Full text reading and applying inclusion and exclusion criteria only 10 articles fulfilled the requirements and could be included in the systematic review.

Round 2

Reviewer 2 Report

The authors have made some of the changes asked. The manuscript can be published in the present form.